# Bibliometric Analysis of Literature on Physical Activity and COVID-19

**DOI:** 10.3390/ijerph19127116

**Published:** 2022-06-10

**Authors:** Apichai Wattanapisit, Manas Kotepui, Sanhapan Wattanapisit, Noah Crampton

**Affiliations:** 1Department of Family and Community Medicine, University of Toronto, Toronto, ON M5G 1V7, Canada; apichai.wattanapisit@mail.utoronto.ca; 2Department of Clinical Medicine, School of Medicine, Walailak University, Tha Sala, Nakhon Si Thammarat 80160, Thailand; 3Department of Medical Technology, School of Allied Health Sciences, Walailak University, Tha Sala, Nakhon Si Thammarat 80160, Thailand; manas.ko@wu.ac.th; 4Primary Care Unit, Thasala Hospital, Tha Sala, Nakhon Si Thammarat 80160, Thailand; th.sanhapan@gmail.com

**Keywords:** bibliometric analysis, COVID-19, physical activity, physical inactivity

## Abstract

The coronavirus disease 2019 (COVID-19) pandemic induced a sudden surge in COVID-19 related publications. This bibliometric analysis aimed to analyze literature on physical activity and COVID-19 published in the PubMed database. The search terms ((physical activity [MeSH Terms] OR physical inactivity [MeSH Terms]) AND COVID-19 [MeSH Terms]) were applied to obtain publications from the inception of PubMed to February 2022. The analyses included the year of publication, type of publication, and origin of publication by country, region, and country income. The research areas were analyzed for research articles and systematic reviews. Of 1268 articles, 143 articles were excluded, and 1125 articles were analyzed. A total of 709 articles (63.02%) were published in 2021. A majority of publications were research articles (*n* = 678, 60.27%). The USA (*n* = 176, 15.64%), countries in the European Region (*n* = 496, 44.09%), and high-income countries (*n* = 861, 76.53%) were dominant publishing countries. Of 699 research articles and systematic reviews, surveillance and trends of physical activity were the main research area, followed by health outcomes, and correlates and determinants of physical activity. There is a wide gap in publication productivity in the field of physical activity and health during the pandemic among different countries’ economic statuses.

## 1. Introduction

The development of research in the field of physical activity and health started in the 1950s [1]. A study on the association between coronary heart disease and physical activity of bus drivers and conductors in London published in 1953 was a milestone in this research field [1,2]. The growth of the research field in the past 70 years (1950–2019) is reflected by a large number of publications (479,712 articles) in the major health (PubMed) and interdisciplinary (Scopus and Web of Science) databases [3]. A search on PubMed using Medical Subject Headings (MeSH) of physical activity on 27 February 2022 found 226,476 articles. The highest number of publications was in 2020 (*n* = 17,666), and the number dropped by 13% to 15,570 in 2021 [4].

Since late 2019, a newly emerging disease, coronavirus disease 2019 (COVID-19), has affected several aspects of health, including physical activity [5,6,7,8,9]. A study by Raynaud et al. showed a rise in COVID-19 related publications and a decrease in the proportion of non-COVID-19 publications in high-impact medical journals [10]. Riccaboni and Verginer analyzed the impacts of COVID-19 on research in the PubMed database and found the COVID-19 pandemic induced a sudden surge in COVID-19 related publications [11]. In contrast, non-COVID-19 research areas experienced a remarkable reduction in overall publishing rates and funding [11].

In addition, the COVID-19 pandemic has changed levels and patterns of physical activity [12,13]. Changes in social and health determinants led to the alteration of physical activity [14,15,16]. These factors may affect the characteristics of and trends in research and publication in this field. To the best of our knowledge, there is a gap in knowledge of understanding of the literature in the field of physical activity and health related to the COVID-19 pandemic. Many studies related to physical activity and COVID-19 were published in journals indexed in PubMed. A first step toward improving research capacity on physical activity is to better understand the change in trends in research published on this subject matter brought on by the COVID-19 pandemic. This study aimed to analyze the characteristics of the literature on physical activity and COVID-19 by determination of the year of publication, type of publication, country of publication, and physical activity and health research areas.

## 2. Materials and Methods

### 2.1. Data Source and Search Strategy

Published articles were retrieved from PubMed, a database of biomedical and life sciences literature that contained more than 33 million records [17]. The search recruited articles from the inception of PubMed to 26 February 2022. The search terms include two key components: physical activity and COVID-19. Physical inactivity was included as a reverse term of physical activity. The MeSH were applied to obtain relevant terms. The Boolean operators “OR” and “AND” were used. Table 1 presents the description of the search strategy using MeSH terms. The search terms were (physical activity [MeSH Terms] OR physical inactivity [MeSH Terms]) AND COVID-19 [MeSH Terms].

### 2.2. Data Management and Extraction

The search results retrieved from PubMed were downloaded as a .csv file to maintain the number of articles on the search date. The .csv file was converted to a .xlsx (a default file format for Microsoft Excel).

The article title and year of publication were kept in their original forms (.csv file). Type of publication was classified as (i) research/original article, (ii) systematic review, (iii) editorial, (iv) letter to the editor/research letter, and (v) miscellaneous article. The country of publication was defined as the country of the first author’s affiliation. The countries were grouped according to the World Health Organization (WHO) regions: (i) African Region, (ii) Eastern Mediterranean Region, (iii) European Region, (iv) Region of the Americas, (v) South-East Asia Region, and (vi) Western Pacific Region [18]. The World Bank’s country classification by income in 2021–2022 was assigned for each country as (i) low income, (ii) lower middle income, (iii) upper middle income, and (iv) high income [19].

The lead author (A.W.) reviewed the titles and abstracts to identify the physical activity research area for research/original articles and systematic reviews (with or without meta-analyses): (i) surveillance and trends, (ii) correlates and determinants, (iii) health outcomes, (iv) interventions and programs, (v) policies [20], and (vi) others (Table 2). Articles that involved more than one research area were classified into the two most relevant areas. Articles were excluded from analysis if their subject matter did not relate to physical activity and health, involved research on animal subjects or in vitro samples, involved exercise diagnostic tests for COVID-19, or were conducted in rehabilitation settings. The data extraction was recorded as spreadsheets in Microsoft Excel (Office 365 University Package; Microsoft, Redmond, WA, USA).

### 2.3. Data Analysis

Each included article was identified as a type of publication and the country of the first author. The region and income of the country were checked on the WHO and the World Bank’s websites [18,19]. Data were presented in frequencies and percentages.

Subsequently, research/original articles and systematic reviews were analyzed qualitatively by the lead author (A.W.) to identify the research areas according to the description in Table 2. The authors (M.K. and S.W.) cross-checked the analysis by adjudicating every 20th article. Subsequently, all authors discussed to resolve any disagreement and finalize the analysis. The research areas were presented as the number of articles in each category. An article that involved two research areas was counted in both areas.

## 3. Results

Of 1268 articles found, 143 articles were excluded. A majority of articles (*n* = 709, 63.02%) were published in 2021, while the rest of articles were published in 2020 (*n* = 341, 30.31%) and 2022 (*n* = 75, 6.67%). Research or original articles were the most common publication type (*n* = 678, 60.27%) (Table 3). The USA (*n* = 176, 15.64%), the European Region (*n* = 496, 44.09%), and high-income countries (*n* = 861, 76.53%) were the most dominant publication productivity (Table 4).

Of 699 articles, 524 articles were classified into a single research area, and 175 articles presented more than one research area. Nearly 60% (*n* = 416 out of 699 articles) of research/original articles and systematic reviews related to COVID-19 were in the area of surveillance and trends. The second most common research area was health outcomes (*n* = 207/699, 29.61%) (Table 5).

## 4. Discussion

A majority of articles related to physical activity and COVID-19 published in the PubMed database were research or original articles (60.27%) and systematic reviews (1.87%). The rest of the publications (37.86%) were classified as non-research articles. Authors affiliated with institutions in the USA, the UK, and Italy contributed more than three in ten publications. More than three-quarters of publications were produced by authors in high-income countries. In contrast, there was no publication from authors in low-income countries. Countries from the European and Americas regions contributed to nearly three-quarters of the publications. Most research articles and systematic reviews focused on surveillance and trends, health outcomes, and correlates and determinants of physical activity during the COVID-19 pandemic.

This study found approximately four in ten publications relevant to physical activity and COVID-19 were published as non-research articles by February 2022. The proportion of non-research articles was higher compared to the ratio of research and non-research articles in other fields [21]. A study by Kotepui and colleagues reported the percentages of non-research articles related to diets and breast cancer research were 18% of publications in Asian countries and 32.4% in other regions of the world [21]. A number of non-research articles constituted an author’s opinions or hypotheses. In addition, some non-research articles may be reviewed by different review processes compared to research articles [22]. This is a consideration of the quality of non-research publications.

Our study presented the top ten dominant countries in physical activity and health publications. Of ten countries, eight are classified as high-income countries (except that Brazil and China are upper-middle-income countries). This finding was consistent with previous evidence. Fontelo et al. analyzed the publication trend in PubMed from 1995 to 2015 and identified the top 30 publishing countries [23]. The top ten countries in our study were in the list of high publication productivity in PubMed. By the WHO’s world regions, compared to before the COVID-19 pandemic (1950 to 2019), our findings showed the highest proportion of publications in the COVID-19 pandemic was produced by the European Region (44.09% vs. 35.1%) followed by the Region of the Americas (30.31% vs. 43.3%), Western Pacific Region (16.53% vs. 14.6%), Eastern Mediterranean Region (5.33% vs. 2.2%), South-East Asia Region (2.49% vs. 2.2%), and African Region (1.24% vs. 2.2%) [3].

In terms of country income, our findings were in line with a study by Ramírez Varela et al. that higher-income countries produced more publications on physical activity (high-income countries: 76.53% vs. 80.2%, upper-middle-income: 17.96% vs. 14.8%, low-middle- income: 5.51% vs. 3.8%, and low-income: 0 vs. 1.1%) and publications per capita from 1950 to 2019 [3]. This reflected similar publication trends during and before the COVID-19 pandemic. The trend that high-income countries contributed to over 80% of publications was found in other medical and health research fields [24,25,26]. A possible hypothesis that may explain the publication productivity of different countries’ economic statuses was the diverse mechanisms to drive research productivity [27]. In addition, physical activity policies were more effective in higher-income countries [28]. The interactions among country income, research productivity, and effectiveness of policies could be explained by the insights of collaboration between scientists and policymakers in high-income countries [29].

We identified the five main areas of physical activity research according to the expert excerpts [20]. However, the five research areas focus on the actions of global public health. Therefore, we added the “others” category for studies that could not be suitably classified into the first five areas. A number of research articles and systematic reviews did not singularly aim to study physical activity; nevertheless, they investigated health behaviors, including physical activity, so we did not exclude those articles. As such, our findings on research areas did not represent the primary outcomes of each article. Studies on policies to improve physical activity during the COVID-19 pandemic were less likely to be published as research articles and systematic reviews. Publications in this area may be published as other article types (e.g., opinions). On the other hand, there was a lack of studies on evidence-based policies regarding physical activity and health during the COVID-19 pandemic.

Our findings revealed some observations and recommendations for scholars and future research. A large proportion of non-research articles were published in the form of expert excerpts. Although these pieces of literature raised valuable knowledge, the knowledge of and perspectives toward COVID-19 were dynamic, especially in the early phase of the outbreak [30,31]. Therefore, these publications should be carefully interpreted. Another finding was that the publication productivity seemed to rely on the country’s income. Compared to a small proportion of publication productivity before the pandemic period, low-income countries had zero output during the pandemic. This finding could reflect that research on physical activity was not the priority during the pandemic in these countries. To improve the global research productivity in physical activity and health, supportive mechanisms and research capacity building should be implemented in low- and middle-income countries. In terms of research knowledge, there was a lack of studies on physical activity-related policies. Future research should strengthen this area to advance the knowledge of physical activity and health.

There were three major strengths of this study. First, the search strategy was inclusive. We used the MeSH terms of both ends of the spectrum of physical activity (i.e., physical activity and physical inactivity) and COVID-19 to obtain the variety of terms used in articles. Second, we excluded non-relevant articles based on the exclusion criteria prior to the analysis. Some published bibliometric analyses included all search results for the analysis. We also excluded some articles that contained keywords relating to physical activity (e.g., running during the storm, swimming against the tide) in their titles [32,33], however, their contents were not related to physical activity. Third, we did not analyze only publication details (i.e., article types, origins of articles). We utilized a conceptual framework published by a group of experts in physical activity and health to identify research areas [20]. These could help us understand the overall trends of publications in the field of physical activity and health during the COVID-19 pandemic.

Despite the strengths of the study, there are certain limitations worth noting. First, we indicated the type of publication based on article types assigned by the journals. However, some journals considered systematic reviews as review articles, while others classified these as research/original articles. We assigned these articles following the Preferred Reporting Items for Systematic Reviews and Meta-Analyses (PRISMA) guideline with/without the systematic review registration (e.g., PROSPERO) as “systematic review” in our analysis. Second, the country of the first author’s affiliation might not represent the site of the study, as the first author may conduct the study in another country or several countries. Our analysis cited only accounts for the affiliating country of the first author. Third, this study did not include the analyses of keywords, author names, and affiliation names.

## 5. Conclusions

This bibliometric analysis disclosed the characteristics and trends of the literature on physical activity and COVID-19 published in PubMed up to February 2022. A majority of articles were research or original articles, while nearly 40 percent were classified as non-research articles (e.g., letter to the editor, editorial, opinions, narrative reviews, case reports). The USA, European countries and high-income countries were dominant publishing countries of publications on physical activity and health during the COVID-19 pandemic. There was no publication by lead authors affiliated with institutions in low-income countries. This reflected the impact of the country’s income on the publication productivity in this field. More than half of the research articles and systematic reviews involved the research area of surveillance and trends of physical activity. Physical activity-related policies were the least common area among research articles and systematic reviews. Building research capacity and supporting the mechanisms to drive research productivity in low- and middle-income countries are required to improve the global research productivity in the field of physical activity and health.

## Figures and Tables

**Table 1 ijerph-19-07116-t001:** Search terms using MeSH terms in PubMed.

Physical Activity		COVID-19 [MeSH Terms]
Physical Activity [MeSH Terms]	Physical Inactivity [MeSH Terms]	
Exercise Exercises Physical Activity Activities, Physical Activity, Physical Physical Activities Exercise, Physical Exercises, Physical Physical Exercise Physical Exercises Acute Exercise Acute Exercises Exercise, Acute Exercises, Acute Exercise, Isometric Exercises, Isometric Isometric Exercises Isometric Exercise Exercise, Aerobic Aerobic Exercise Aerobic Exercises Exercises, Aerobic Exercise Training Exercise Trainings Training, Exercise Trainings, Exercise	Sedentary Behavior Behavior, Sedentary Sedentary Behaviors Sedentary Lifestyle Lifestyle, Sedentary Physical Inactivity Inactivity, Physical Lack of Physical Activity Sedentary Time Sedentary Times Time, Sedentary	COVID-19 COVID-19 SARS-CoV-2 Infection Infection, SARS-CoV-2 SARS-CoV-2 Infection SARS-CoV-2 Infections 2019 Novel Coronavirus Disease 2019 Novel Coronavirus Infection 2019-nCoV Disease 2019-nCoV Disease 2019-nCoV Diseases Disease, 2019-nCoV COVID-19 Virus Infection COVID-19 Virus Infection COVID-19 Virus Infections Infection, COVID-19 Virus Virus Infection, COVID-19 Coronavirus Disease 2019 Disease 2019, Coronavirus Coronavirus Disease-19 Coronavirus Disease-19 Severe Acute Respiratory Syndrome Coronavirus 2 Infection SARS Coronavirus 2 Infection COVID-19 Virus Disease COVID-19 Virus Disease COVID-19 Virus Diseases Disease, COVID-19 Virus Virus Disease, COVID-19 2019-nCoV Infection 2019-nCoV Infection 2019-nCoV Infections Infection, 2019-nCoV COVID19 COVID-19 Pandemic COVID-19 Pandemic Pandemic, COVID-19 COVID-19 Pandemics
*n* = 226,476	*n* = 12,203	*n* = 142,054

**Table 2 ijerph-19-07116-t002:** Research area description.

Research Area	Description
Surveillance and trends	Prevalence and/or characteristics of PA (including exercise, movement), sedentary behavior (including physical inactivity, sitting time, screen time) during the COVID-19 pandemic.
Correlates and determinants	Demographic, social, behavioral, health, and external factors related to PA (including exercise, movement), sedentary behavior (including physical inactivity, sitting time, screen time). Not include a direct effect of COVID-19 and policies related to COVID-19 (e.g., lockdown).
Health outcomes	Effects of PA (including exercise, movement), sedentary behavior (including physical inactivity, sitting time, screen time) on health outcomes, including, medical conditions, lab results, physical and mental health, well-being, quality of life, and health behaviors.
Interventions and programs	Interventions, programs, recommendations, and guidelines that aim to increase PA (including exercise, movement), and reduce sedentary behavior (including physical inactivity, sitting time, screen time) or using PA and sedentary behavior as a medium to improve health outcomes.
Policies	PA/sedentary behavior-related policies, laws, and regulations that focus on implementation, practice-based opportunities, working across disciplines/sectors.
Others	Other areas that are unable to classify into the above areas (e.g., knowledge, skills, and attitude studies, indirect effects of PA on physiological changes, instrument validation).

**Table 3 ijerph-19-07116-t003:** Type of publication.

Type	Number (*n* = 1125)	Percentage
Research/original article	678	60.27
Letter to the editor/research letter	87	7.73
Editorial	41	3.64
Systematic review	21	1.87
Miscellaneous article *	298	26.49

***** Miscellaneous articles included other types of review articles (e.g., narrative reviews, scoping reviews), opinion/perspective/commentary/communication articles, case reports/series, brief reports, research reports, and study protocol articles.

**Table 4 ijerph-19-07116-t004:** Country of publication.

Country/Income/Region	Number (*n* = 1125)	Percentage
USA	176	15.64
UK	98	8.71
Italy	93	8.27
Brazil	84	7.47
Spain	81	7.20
China	54	4.80
Canada	53	4.71
Japan	44	3.91
Australia	40	3.56
Poland	30	2.67
Others (64 countries/territories)	372	33.07
European Region	496	44.09
Region of the Americas	341	30.31
Western Pacific Region	186	16.53
Eastern Mediterranean Region	60	5.33
South-East Asia Region	28	2.49
African Region	14	1.24
High income	861	76.53
Upper middle income	202	17.96
Lower middle income	62	5.51
Low income	0	0

**Table 5 ijerph-19-07116-t005:** Physical activity research area extracted from research/original articles and systematic reviews.

Research Area *	Surveillance and Trends	Correlates and Determinants	Health Outcomes	Interventions and Programs	Policies	Others
**Surveillance and trends**	247	91	71	0	0	0
**Correlates and determinants**	6	50	5	0	0	0
**Health outcomes**	1	1	129	0	0	0
**Interventions and programs**	0	0	0	60	0	0
**Policies**	0	0	0	0	5	0
**Others**	0	0	0	0	0	33

* The intersection of the similar research areas by row and column means the articles presented only a single research area. Otherwise, the articles involved two research areas.

## Data Availability

Not applicable.

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
