# Peer review of "Bibliometric Analysis of Literature on Physical Activity and COVID-19"

_ijerph, 2022, doi:10.3390/ijerph19127116_

Round 1

Reviewer 1 Report

The authors propose a bibliometric analysis on Physical Activity and COVID-19. The subject is interesting and important for the area. However, despite this interesting idea, the development is weak. The main research problem is not highlighted. Moreover, bibliometric studies currently analyze in depth and in details the literature data, which is not performed by the authors' study. In fact, the result is poor and easily obtained without much effort from the bibliographic database platforms themselves. In this way, I suggest the authors study the current bibliometric analyzes on Covid-19, the current methods that have been used so that they can carry out a new, more investigative research with much more data and details about the problem.

Author Response

Reviewer 1

The authors propose a bibliometric analysis on Physical Activity and COVID-19. The subject is interesting and important for the area. However, despite this interesting idea, the development is weak.

Thank you for your helpful comments.

The main research problem is not highlighted.

Thank you for the comment. Another reviewer (reviewer#2) also addressed the similar issue. We have added the research gap in the introduction (paragraph 3).

“In addition, the COVID-19 pandemic has changed levels and patterns of physical activity [12,13]. Changes in social and health determinants led to the alteration of physical activity [14-16]. These factors may affect the characteristics of and trends in research and publication in this field. To the best of our knowledge, there is a gap in knowledge of understanding of literature in the field of physical activity and health related to the COVID-19 pandemic. Many studies related to physical activity and COVID-19 were published in journals indexed in PubMed. A first step toward improving research capacity on physical activity is to better understand the change in trends in research publication on this subject matter brought on by the COVID-19 pandemic. This study aimed to analyze the characteristics of the literature on physical activity and COVID-19 through determination of the year of publication, type of publication, country of publication, and physical activity and health research areas.”

Moreover, bibliometric studies currently analyze in depth and in details the literature data, which is not performed by the authors' study. In fact, the result is poor and easily obtained without much effort from the bibliographic database platforms themselves. In this way, I suggest the authors study the current bibliometric analyzes on Covid-19, the current methods that have been used so that they can carry out a new, more investigative research with much more data and details about the problem.

We agree with your comments. This was the main issue that we discussed before the data extraction and analysis. Please allow us to explain. We reviewed a number of published bibliometric studies and realized that these studies vary in methods. We did not find a standard reporting guideline for bibliometric studies on the Equator Network website (https://www.equator-network.org/).

Many studies imported all search results from database(s) (without exclusion) to a software (e.g., VOSviewer https://www.vosviewer.com/). The software could automatically analyze several indicators (e.g., authors, affiliations, keywords) and generate visualizing bibliometric networks. This software is widely used in published articles.   

We imported the search results to the software. The software generated a nice visualizing bibliometric networks for keywords, however, we found some inconsistency (i.e., *covid-19, *sars-cov-2 (with asterisk signs) VS. covid-19, sars-cov-2 in a visualizing map. In addition, we realized that some articles contained keywords relating to physical activity (e.g., running during the storm, swimming against the tide) to express their titles, however, their contents were not related to physical activity. Therefore, we decided to review all titles and abstract manually instead of including all articles.

To strengthen our analysis, we decided to utilize a conceptual framework published by a group of experts in physical activity and health to highlight the characteristics and trends of publications. We have explained this as a strength of our study.

“There were three major strengths of this study. First, the search strategy was inclusive. We use the MeSH terms of both ends of the spectrum of physical activity (i.e., physical activity and physical inactivity) and COVID-19 to obtain the variety of terms used in articles. Second, we excluded non-relevant articles based on the exclusion criteria prior to the analysis. Some published bibliometric analyses included all search results for the analysis. We also excluded some articles contained keywords relating to physical activity (e.g., running during the storm, swimming against the tide) to express their titles [32,33], however, their contents were not related to physical activity. Third, we did not analyze only publication details (i.e., article types, origins of articles). We utilized a conceptual framework published by a group of experts in physical activity and health to identify research areas [20]. These could help understand the overall trends of publications in the field of physical activity and health during the COVID-19 pandemic.”

Although analyses of author names, affiliation names, and keywords were helpful to understand the bibliometric analysis (several previous publications included these), these variables were not the main outcomes of our study. However, we believe that our analysis of research areas (a strength of this study) is more useful for our study compared to the analyses of author names, affiliation names, and keywords. We have added this as our limitation.

“Third, this study did not include the analyses of keywords, author names, and affiliation names.”   

Reviewer 2 Report

This article used bibliometric analytical methodology to analyze the characteristics of the literature on physical activity and COVID-19. While I appreciate the massive work done by the research team, the contribution of this article is not strong. The research team should consider making an effort to revise the discussion and introduction section to clearly articulate the contribution of this article.   

Introduction

  • The research gap is not explicit/clear enough

Materials and Methods

Regarding 2.2 data management and analysis section. The data extraction part was clearly written but data analysis was too short. Suggest to describe data analysis in a separate paragraph.

Results

Results are clearly presented.

Discussion

The discussion section is disappointing in a sense that it fails to discuss how the trends and characteristics identified from the bibliometric analysis could help physical activity scholar/ health science scholar to make a judgement on what is needed for future research. The contribution of this article is not strong. The current strengths identified by authors are not convincing enough.

Author Response

Reviewer 2

This article used bibliometric analytical methodology to analyze the characteristics of the literature on physical activity and COVID-19. While I appreciate the massive work done by the research team, the contribution of this article is not strong. The research team should consider making an effort to revise the discussion and introduction section to clearly articulate the contribution of this article.  

Thank you for your review and the helpful comments.

Introduction

The research gap is not explicit/clear enough

We have added some statements and references to indicate the research gap in knowledge. These additional contents will bridge the gap between the previous paragraph and the final paragraph.

“In addition, the COVID-19 pandemic has changed levels and patterns of physical activity [12,13]. Changes in social and health determinants led to the alteration of physical activity [14-16]. These factors may affect the characteristics of and trends in research and publication in this field. To the best of our knowledge, there is a gap in knowledge of understanding of literature in the field of physical activity and health related to the COVID-19 pandemic. Many studies related to physical activity and COVID-19 were published in journals indexed in PubMed. A first step toward improving research capacity on physical activity is to better understand the change in trends in research publication on this subject matter brought on by the COVID-19 pandemic. This study aimed to analyze the characteristics of the literature on physical activity and COVID-19 through determination of the year of publication, type of publication, country of publication, and physical activity and health research areas.”  

Materials and Methods

Regarding 2.2 data management and analysis section. The data extraction part was clearly written but data analysis was too short. Suggest to describe data analysis in a separate paragraph.

We have added a subsection (2.3. Data Analysis) to describe our data analysis.

“2.3. Data Analysis 

Each included article was identified a type of publication and a country of the first author. The region and income of the country were checked on the WHO and the World Bank’s websites [13,14]. Data were presented in frequencies and percentages.

Subsequently, research/original articles and systematic reviews were analyzed qualitatively by the lead author (A.W.) to identify the research areas according to the description in Table 2. The authors (M.K. and S.W.) cross-checked the analysis by adjudicating every 20th articles. Subsequently, all authors discussed to resolve any disagreement and finalize the analysis. The research areas were presented as the number of articles in each category. An article that involved two research areas were counted in both areas.”

Results

Results are clearly presented.

Thank you for the comment.

Discussion

The discussion section is disappointing in a sense that it fails to discuss how the trends and characteristics identified from the bibliometric analysis could help physical activity scholar/ health science scholar to make a judgement on what is needed for future research. The contribution of this article is not strong. The current strengths identified by authors are not convincing enough.

We have inserted a paragraph to present observations and recommendations according to our findings.

“Our findings revealed some observations and recommendations for scholars and future research. A large proportion of non-research articles were published in the forms of expert excerpts. Although these pieces of literature raised the valuable knowledge, the knowledge of and perspectives toward COVID-19 were dynamic, especially, in the early phase of the outbreak [30,31]. Therefore, these publications should be carefully interpreted. Another finding was the publication productivity seemed to rely on country’s income. Compared to a small proportion of publication productivity before pan-demic period, low-income countries had a zero output during the pandemic. This finding could reflect that research on physical activity was not the priority during the pandemic among these countries. To improve the global research productivity in physical activity and health, supportive mechanisms and research capacity building should be implemented in low- and middle-income countries. In terms of research knowledge, there was a lack of studies on physical activity related policies. Future re-search should strengthen this lacking area to advance the knowledge of physical activity and health.”

We have revised the strengths of the study.  

“There were three major strengths of this study. First, the search strategy was inclusive. We use the MeSH terms of both ends of the spectrum of physical activity (i.e., physical activity and physical inactivity) and COVID-19 to obtain the variety of terms used in articles. Second, we excluded non-relevant articles based on the exclusion criteria prior to the analysis. Some published bibliometric analyses included all search results for the analysis. We also excluded some articles contained keywords relating to physical activity (e.g., running during the storm, swimming against the tide) to express their titles [32,33], however, their contents were not related to physical activity. Third, we did not analyze only publication details (i.e., article types, origins of articles). We utilized a conceptual framework published by a group of experts in physical activity and health to identify research areas [20]. These could help understand the overall trends of publications in the field of physical activity and health during the COVID-19 pandemic.”

Reviewer 3 Report

Thank you very much for reading this text. The text is certainly valuable, but it requires supplements and clarification of individual threads.

The article is methodologically and technically well organized. The research is exploratory. A typical review article, but carefully prepared. It raises an important topic of physical activity during COVID-19.

This bibliometric analysis was aimed at analyzing the literature on physical activity and COVID-19 published in PubMed. The terms searched by the authors were used to obtain publications from the inception of PubMed until February 2022. The analyzes covered the year of publication, type of publication, and publication origin by country, region, and country income. Research areas were analyzed for scientific articles and systematic reviews.

Among nearly 700 scientific articles and systematic reviews, the main research area was supervision and trends in physical activity, followed by health outcomes, and correlates and determinants of physical activity. It has been found that during a pandemic, there is a large difference in the publication efficiency of physical activity and health between different economic statuses of countries.

The only aspect that could be a little more detailed is the conclusions of the hundreds of articles analyzed, which are limited to a few lines. It is suggested to extend this section of the article.

Author Response

Reviewer 3

Thank you very much for reading this text. The text is certainly valuable, but it requires supplements and clarification of individual threads.

Thank you for your review and suggestions.

The article is methodologically and technically well organized. The research is exploratory. A typical review article, but carefully prepared. It raises an important topic of physical activity during COVID-19.

This bibliometric analysis was aimed at analyzing the literature on physical activity and COVID-19 published in PubMed. The terms searched by the authors were used to obtain publications from the inception of PubMed until February 2022. The analyzes covered the year of publication, type of publication, and publication origin by country, region, and country income. Research areas were analyzed for scientific articles and systematic reviews.

Among nearly 700 scientific articles and systematic reviews, the main research area was supervision and trends in physical activity, followed by health outcomes, and correlates and determinants of physical activity. It has been found that during a pandemic, there is a large difference in the publication efficiency of physical activity and health between different economic statuses of countries.

Thank you.

The only aspect that could be a little more detailed is the conclusions of the hundreds of articles analyzed, which are limited to a few lines. It is suggested to extend this section of the article.

Thank you for the comment. We have extended this section by inserting brief description of the study, the key findings, and future recommendations.

“This bibliometric analysis disclosed the characteristics and trends of literature on physical activity and COVID-19 published in PubMed up to February 2022. A majority of articles were research or original articles, while nearly 40 percent were classified as non-research articles (e.g., letter to the editor, editorial, opinions, narrative reviews, case reports). The USA, European countries, and high-income countries were dominant publishing countries of publications on physical activity and health during the COVID-19 pandemic. There was no publication by lead authors affiliated with institutions in low-income countries. This reflected the impact of the country’s income on the publication productivity in this field. More than half of research articles and systematic reviews involved the research area of surveillance and trends of physical activity. Physical activity related policies were the least common area among research articles and systematic reviews. Building research capacity and supporting the mechanisms to drive research productivity in low- and middle-income countries are required to improve the global research productivity in the field of physical activity and health.”

Round 2

Reviewer 2 Report

Thank you for the revision.

I have no further comment. 

Author Response

Dear Reviewer, 

We wish to thank you for your suggestions that help improve our manuscript.